# Neurochemical Changes in the Brain and Neuropsychiatric Symptoms in Clinically Isolated Syndrome

**DOI:** 10.3390/jcm9123909

**Published:** 2020-12-02

**Authors:** Wojciech Guenter, Maciej Bieliński, Robert Bonek, Alina Borkowska

**Affiliations:** 1Department of Clinical Neuropsychology, Nicolaus Copernicus University, 87-100 Toruń, Poland, and Collegium Medicum, 85-094 Bydgoszcz, Poland; bielinskim@gmail.com (M.B.); alab@cm.umk.pl (A.B.); 2Regional Specialized Hospital in Grudziądz, Division of Neurology and Clinical Neuroimmunology, 86-300 Grudziądz, Poland; bonek.robert@gmail.com

**Keywords:** clinically isolated syndrome, multiple sclerosis, cognition, neuropsychology, proton magnetic resonance spectroscopy

## Abstract

To assess cognitive impairment and affective symptoms and their association with damage to normal-appearing white matter (NAWM) in patients with clinically isolated syndrome (CIS), we compared neuropsychological test scores between patients with CIS and healthy controls and examined correlations between these and proton magnetic resonance spectroscopy (^1^H-MRS) outcomes in patients with CIS. Forty patients with CIS and 40 healthy participants were tested with a set of neuropsychological tests, which included the Beck Depression Inventory (BDI) and the Hospital Anxiety and Depression Scale (HADS). Single-voxel ^1^H-MRS was performed on frontal and parietal NAWM of patients with CIS to assess ratios of *N*-acetyl-aspartate (NAA) to creatine (Cr), *myo*-inositol (mI), and choline (Cho), as well as mI/Cr and Cho/Cr ratios. Patients with CIS had lower cognitive performance and higher scores for the BDI and anxiety subscale of HADS than healthy controls. There were significant correlations between the following neuropsychological tests and metabolic ratios in the frontal NAWM: Stroop Color-Word Test and Cho/Cr, Symbol Digit Modalities Test and mI/Cr, as well as NAA/mI, Go/no-go reaction time, and NAA/Cho, as well as NAA/mI, Californian Verbal Learning Test, and NAA/Cr. BDI scores were related to frontal NAA/mI and parietal NAA/Cr and Cho/Cr ratios, whereas HADS-depression scores were associated with frontal NAA/Cr and NAA/mI and parietal NAA/Cr and Cho/Cr ratios. HADS-anxiety correlated with parietal NAA/Cr ratio. This study suggests that neurochemical changes in the NAWM assessed with single-voxel ^1^H-MRS are associated with cognitive performance and affective symptoms in patients with CIS.

## 1. Introduction

Cognitive impairment occurs in 18–57% of patients with clinically isolated syndrome (CIS) [1,2]. Although cognitive functioning is not associated with structural brain changes such as T2- and T1-weighted lesion burden [3,4] and brain atrophy [5,6], damage to the normal-appearing brain tissue (NABT) assessed with magnetization transfer imaging and diffusion tensor imaging (DTI) has been linked to cognitive impairment in CIS [1,7]. Proton magnetic resonance spectroscopy (^1^H-MRS) also reveals damage to the NABT in the early stages of CIS and multiple sclerosis (MS) [8,9]. These neurochemical changes are associated with cognitive impairment in patients with MS [10,11], but this relation has not been evaluated in patients with CIS.

Patients with CIS also have an increased prevalence of affective symptoms [4,12], which are related to structural brain changes in patients with MS and CIS [13,14]. A study on patients with clinically defined MS indicated that DTI alterations in the NABT are associated with depression symptoms [15]. However, we do not know if neurochemical changes in NABT contribute to these symptoms in patients with CIS.

Neurochemical changes in the brain corresponding with damage to the NABT include: decreased concentration of NAA or NAA/Cr ratio, indicating neuronal and axonal loss or dysfunction; increased concentration of Cho or Cho/Cr ratio, corresponding with increased membrane turnover (synthesis and degradation); myelin damage and repair processes; and increased concentration of mI or mI/Cr ratio, which is associated with elevated numbers and activity of glial cells [16]. NAA/mI and NAA/Cho might increase sensitivity for mild damage to brain tissue because such combined ratios reflect a broader aspect of the pathology [17,18].

The aim of this study was to assess cognitive performance and affective symptoms in patients with CIS and to evaluate if damage to the normal-appearing white matter (NAWM) assessed with ^1^H-MRS is associated with cognitive impairment and affective symptoms in these patients.

## 2. Methods

### 2.1. Participants and Procedures

Forty patients with CIS, according to the definition by Miller et al. [19], hospitalized in the Division of Neurology and Clinical Neuroimmunology at the Regional Specialized Hospital in Grudziądz, Poland, were studied between 2014 and 2016. The study was prospective. Patients who had a single neurological episode suggesting an inflammatory demyelinating cause after the appropriate differential diagnosis between 2 months and 4 years prior to enrollment were recruited for the study. Patients with other neurologic, systemic, or major psychiatric disease; drug or alcohol addiction; who had an upper limb or visual disturbance; or used a drug that would interfere with neuropsychological testing were excluded. Patients who participated in the study were neurologically stable and had not taken steroids for at least 1 month before assessment. None of the participating patients had been treated with a disease-modifying therapy. A new revision by MAGNIMS of the McDonald 2010 criteria were published in 2016 [20], so we retrospectively used these revised criteria to define which study participants fulfilled the criteria for MS. Forty demographically matched healthy volunteers were recruited as healthy participant (HP) controls.

Physical and neurological examinations, including the Expanded Disability Status Scale (EDSS), were performed on all participants (healthy controls were normal upon neurologic examination) before cognitive function and affective symptoms were evaluated. An MRI of the brain and cervical spine, including ^1^H-MRS, was then performed for all patients. These procedures were performed within 5 days.

The study was approved by the Ethics Committee of the Collegium Medicum of Nicolaus Copernicus University, Bydgoszcz, Poland (KB 41/2014 and KB 78/2016). Written informed consent was obtained from all participants.

### 2.2. Clinical and Neuropsychological Assessments

Depression symptoms were evaluated with the Polish adaptation of the Beck Depression Inventory (BDI) [21]; a total score of 13 was used to indicate the presence of depression symptoms. The Polish version of the Hospital Anxiety and Depression Scale (HADS) [22] was used to assess depression and anxiety symptoms; a cutoff level of 8 points for each subscale was used to indicate the presence of depression or anxiety symptoms.

Neuropsychological assessment included the following tests:Trail Making Test (TMT). Two parts of the TMT from the Halstead-Reitan battery [23] were administered as follows: TMT A with numbers and TMT B with numbers and letters. The test score was the time to complete each task.Stroop Color-Word Test (SCWT). Two stimuli were used for the SCWT [24] as follows: a card with the names of colors written in black ink and a card with the names of colors written in an incongruent color. The measures of performance were the time to name the colors written in ink of an incongruent color and the number of errors made.Symbol Digit Modalities Test (SDMT) [25] is a simple substitution task. The written version of the SDMT was used. The test score was the number of correctly paired numbers with given geometric figures in 90 s.Verbal Fluency Test. In the Verbal Fluency Test [26], participants are asked to generate words beginning with each of the letters F, A, and S, excluding proper nouns (1 min for each letter). The measure of performance was the total number of words generated, excluding repetitions.Neurotest. Participants were administered the Single Reaction Time (SRT) test and the Go/no-go task of the Polish version of Neurotest, a computer neuropsychological test battery comprising 11 tools. In the SRT task, the participant is asked to press a computer key as quickly as possible when a green circle appears on the screen (the stimulus appears 25 times). The measure of performance is the average response time. In the Go/no-go task, the participant is told to react when a green square appears on the screen (“go” reaction) and not to react when a blue square appears (“no-go” reaction). The stimuli appear in a random order (75 green squares and 25 blue squares). The test outcomes are the response time for correct “go” reactions and the number of incorrect “no-go” reactions (indicating perseverative errors).California Verbal Learning Test (CVLT). The Polish version of the CVLT [27] was used to assess verbal memory. The following outcomes were evaluated: number of words immediately recalled from the first list of 16 words (list A) in the 1st and 5th trials, total number of words immediately recalled from the 1st trial to the 5th trial, short delayed free recall (after learning interference list of words—list B), and long delayed free recall (recalling the words from list A after 20 min).

### 2.3. MR Examination and Analysis

Cranial MRI examinations were performed with a 1.5 T Philips Achieva scanner (Philips Medical Systems, Best, The Netherlands) in the Division of Radiology and Diagnostic Imaging at Regional Specialized Hospital in Grudziądz, Poland within 5 days of the neurological and neuropsychological assessments. The maximal slew rate was 80 mT/m/ms, with a maximum gradient strength of 33 mT/m. The brain MRI protocol included sagittal and axial T2-weighted turbo spin echo (TSE) (repetition time [TR], 4847 ms; echo time [TE], 100 ms; turbo factor [TF], 15; slice thickness [ST], 5 mm), axial T1-weighted SE (TR, 450 ms; TE, 15 ms; ST, 5 mm), sagittal three-dimensional (3D) T1 fast field echo (FFE) (TR, 7.1 ms; TE, 3.2 ms; TF, 232; inversion time, 852.8 ms; ST, 1 mm), sagittal 3D fluid-attenuated inversion recovery (FLAIR) (TR, 4800 ms; TE, 315 ms; TF, 178; inversion time, 1660 ms; ST, 0.56 mm), and axial diffusion-weighted imaging (TR, 3668 ms; TE, 100 ms; ST, 5 mm). Axial T1-weighted SE and sagittal 3DT1 FFE were performed before and after the administration of a gadolinium-based contrast agent (0.1 mmol/kg). The cervical spine MRI protocol included coronal and sagittal T2-weighted TSE (TR, 3000 ms; TE, 120 ms; TF, 50; ST, 3 mm), axial T2-weighted TSE (TR, 3700 ms; TE, 110 ms; TF, 33; ST, 3.3 mm), sagittal T2-weighted spectral-attenuated inversion recovery (SPAIR) (TR, 3000 ms; TE, 90 ms; TF, 33; ST, 3 mm), as well as sagittal T1-weighted TSE (TR, 400 ms; TE, 7.8 ms; TF, 5; ST, 3 mm) and axial T1-weighted TSE (TR, 556 ms; TE, 7 ms; TF, 7; ST, 3.3 mm) performed before and after the administration of a gadolinium-based contrast agent. MR images were assessed by an experienced neuroradiologist. The presence and number of lesions suggesting demyelination, including gadolinium-enhanced lesions, were evaluated, as well as their localization (subcortical, periventricular, or infratentorial). The width of the third ventricle was measured using transverse slices from 3D FLAIR. For this, a line was drawn through the long axis of the third ventricle in the section where the ventricle was most visible. Then, a perpendicular line was drawn at the midpoint between the lateral walls of the third ventricle, for which the length was recorded as the width of the third ventricle [28].

^1^H-MRS was acquired at the same time as conventional MRI with the same scanner before the gadolinium-based contrast agent was administered. For planning and locating the target voxel, the following scans were acquired: transverse T2 TSE (TR, 4847 ms; TE, 100 ms; ST, 5 mm), axial T1 (TR, 135 ms; TE, 1.79 ms; ST, 5 mm), and coronal FLAIR (TR, 600 ms; TE, 120 ms; inversion time, 2000 ms). Then, a single-voxel ^1^H-MRS was acquired for the following two volumes of interest (VOIs; 2.7 mL) in NAWM in the left hemisphere: (i) the frontal region anterolaterally to the frontal horn of the lateral ventricle; and (ii) the parietal region superolateral to the trigone of the lateral ventricle (Figure 1). The VOI was defined manually using scans in the transverse, axial, and coronal planes to maximize the area of NAWM sampled while excluding cerebrospinal fluid, gray matter, and demyelinated lesions. In cases where the VOI could not be placed in the left hemisphere because of lesions, the right hemisphere was chosen as there is no evidence for hemispheric differences in the NAWM ^1^H-MRS measures [29,30]. A point-resolved spectroscopy sequence with TR of 2000 ms and TE of 31 and 144 ms was used in the ^1^H-MRS acquisition. Water was suppressed with the chemical shift selective saturation technique (CHESS). Water suppression and shimming were optimized automatically. A medical physicist experienced in ^1^H-MRS analysis performed postprocessing with SpectroView software (Philips). We semi-quantitatively assessed metabolite concentrations by calculating relative peak area ratios for *N*-acetyl-aspartate (NAA)/creatine (Cr) and NAA/choline (Cho) for long TE (144 ms) and Cho/Cr, *myo*-inositol (mI)/Cr, and NAA/mI for short TE (31 ms).

### 2.4. Statistical Analysis

Statistica 13.3 software (StatSoft) was used for all statistical analyses except for calculations of effect sizes. *p*-values of <0.05 were considered statistically significant. The normality of the distribution of the variables was verified using the Shapiro–Wilk test. The homogeneity of variances in compared groups was evaluated with Levene’s test. The arithmetic means and standard deviations (SDs) are shown as measures of central tendency and dispersion for variables with normal distributions. Otherwise, the medians and 25–75th percentiles (Q1–Q3) are shown. Student’s *t*-tests were used to compare quantitative variables between two independent samples, provided that the variables had normal distributions and the variances were homogeneous; otherwise, a Welch’s *t*-test was used. Mann–Whitney *U*-tests were used to compare two groups with quantitative variables that were not normally distributed. A paired sample *t*-test was used to compare dependent samples and a chi-square test was used to compare nominal variables between two groups. To evaluate two qualitative independent variables as predictors of a dependent variable, a two-way analysis of variance was used. Effect sizes were calculated as described previously [31]. Effect sizes for *t*-tests were calculated with Cohen’s *d*. If variances in the compared groups were inhomogeneous, Glass’ Δ was used. Cohen’s *d* for Mann–Whitney *U*-tests was calculated from η^2^, which was calculated from the *U*-statistic [31]. Correlations between two variables were assessed with Pearson’s correlation coefficient (*r*) if variables were normally distributed or the Spearman’s rank correlation coefficient (*R*) if variables were not normally distributed. A multiple regression model was used to calculate the effects of more than one independent variable (predictor) on a dependent variable. In the case of correlation of numerous parameters, the multiple testing procedure (Bonferroni correction) was also performed, re-qualifying the significance of the p coefficient.

## 3. Results

The main characteristics of patients with CIS and HPs are detailed in Table 1. One of the patients with CIS did not perform a short delayed free recall trial in the CVLT. mI/Cr and NAA/mI ratios were not obtained in the frontal VOI in one of the study participants and in the parietal VOI in one other patient.

### 3.1. Neuropsychological Performance in CIS and HP Groups and Associations with Demographic Factors and BDI/HADS and EDSS Scores

Patients with CIS performed worse on the TMT B and SDMT and had longer SCWT times and worse short delayed free recall (SDR) scores on the CVLT than HPs (Table 2). The use of Bonferroni correction in multiple testing changed the value for significance to *p*~0.004, thus only the difference in SDMT performance between patients with CIS and HPs remained significant. Patients with CIS also had higher BDI and HADS-anxiety (HADS-A) scores (Table 2). After Bonferroni correction, only higher HADS-A confirmed statistical significance. Overall, 22.5% and 15% of patients with CIS had symptoms of depression compared with only 5% and 2.5% of HPs according to the BDI and HADS, respectively; 35% of patients with CIS and 22.5% of HPs exhibited signs of anxiety according to the HADS.

Affective symptom severity was correlated with cognitive performance in the CIS group, specifically between SDMT and BDI (*R* = −0.32, *p* = 0.04), SDMT and HADS-A (*R* = −0.33, *p* = 0.04), Go/no-go errors and BDI (*R* = 0.39, *p* = 0.01), and CVLT SDR and HADS-depression (HADS-D) (*R* = −0.34, *p* = 0.04). By contrast, no correlations were observed for the HP group. When the BDI or HADS score was included as a covariate in an analysis of cognitive performance between CIS and HP groups (using multiple linear regression), the difference in CVLT SDR was not significant. However, the differences in SDMT (*p* < 0.00001), TMT B (*p* = 0.02), and SCWT time (*p* = 0.04) remained statistically significant.

Neurological status according to the EDSS was not related to scores on any cognitive test. However, the duration of the disease (time from CIS onset) was associated with SCWT errors (*R* = 0.38, *p* = 0.02).

### 3.2. Classic MRI Findings

Two of the CIS patients (5%) had no lesions according to T2-weighted and FLAIR images. Twenty-seven patients (67.5%) and eight patients (20%) met the Magnetic Resonance Imaging in Multiple Sclerosis (MAGNIMS) 2016 MRI criteria for dissemination in space (DIS) and for diagnosis of MS [20], respectively, whereas 23 patients (57.5%) fulfilled the Barkhof/Tintoré MRI criteria for DIS [32]. Lesions in the cervical spine were present in 19 patients with CIS (47.5%). Gadolinium-enhanced lesions were detected in eight patients (20%) and all of them fulfilled MAGNIMS 2016 MRI criteria for MS. The median width of the third ventricle was 4.25 mm (Q1–Q3, 3.6–4.9 mm).

We tested the hypothesis that patients meeting the DIS criteria will be burdened with more pronounced neuropsychiatric sequelae of the disease, but it has turned out that patients who fulfilled the MRI criteria for DIS (MAGNIMS 2016 or Barkhof/Tintoré MRI criteria) did not differ from those without DIS in BDI, HADS, and cognitive test scores. While verifying whether the presence of gadolinium-enhanced lesions was related to cognitive decline, it was determined that patients with these lesions made more no-go mistakes in the Go/no-go task (median, 5.5 (Q1–Q3, 3.5–8.5) vs. median, 2.5 (Q1–Q3, 1.0–4.0); *d* = 0.86; *p* = 0.01). It was also necessary to determine whether the parameter of central atrophy in the study population was related to the intensity of the neuropsychiatric parameters under study. The width of the third ventricle was found to not be related to BDI or HADS score or to cognitive status. However, it was related to the age of participants (*R* = 0.33, *p* = 0.04). Collectively, our data indicate that only presence of gadolinium-enhanced lesions are correlated with some cognitive parameters.

### 3.3. H-MRS Findings

The means of ^1^H-MRS ratios in the frontal and parietal VOIs are detailed in Table 3. NAA/Cr and NAA/Cho ratios were significantly lower in frontal VOIs than in parietal VOIs. In addition, there were significant linear correlations for NAA/Cr (*r* = 0.33, *p* = 0.04), mI/Cr (*r* = 0.39, *p* = 0.01), Cho/Cr (*r* = 0.47, *p* = 0.002), NAA/Cho (*r* = 0.57, *P* = 0.001), and NAA/mI (*r* = 0.55, *p* = 0.001) ratios between the frontal and parietal VOIs. The width of the third ventricle was correlated only with the frontal NAA/mI ratio (*R* = −0.33, *p* = 0.04).

An analysis was performed whether 1H-MRS differentiates the studied population depending on the specificity of clinical symptoms of CIS. It was found that ^1^H-MRS ratios were not associated with patient age or disease duration, with similar ratios observed among patients with optic neuritis, and other clinical manifestations of CIS. Patients fulfilling MAGNIMS 2016 MRI criteria for MS had a lower parietal NAA/Cho ratio (1.57 ± 0.21 vs. 1.83 ± 0.25, *p* = 0.01). Further analysis showed that EDSS values were correlated with frontal Cho/Cr (*R* = −0.4, *p* = 0.01) and parietal NAA/Cr (*R* = −0.46, *p* = 0.003) and NAA/Cho (*R* = −0.33, *p* = 0.04) ratios. Then, analyzing the correlation with the severity of symptoms of depression and anxiety, significant associations were found between BDI scores and frontal NAA/mI ratios (*R* = −0.35, *p* = 0.03), as well as parietal NAA/Cr (*R* = −0.44, *p* = 0.005) and Cho/Cr (*R* = −0.41, *p* = 0.009) ratios. HADS-D values were correlated with frontal NAA/Cr (*R* = −0.35, *p* = 0.03) and NAA/mI (*R* = −0.33, *p* = 0.04) ratios, as well as parietal NAA/Cr (*R* = −0.47, *p* = 0.002) and Cho/Cr (*R* = −0.34, *p* = 0.03) ratios, whereas HADS-A values correlated only with parietal NAA/Cr ratios (*R* = −0.37, *p* = 0.02). The applied Bonferroni correction confirmed the significance for associations between BDI and parietal NAA/Cr and Cho/Cr ratios, as well as between the HADS-D and parietal NAA/Cr ratio.

In the next step, an analysis of connections with the results of cognitive tests was performed. ^1^H-MRS ratios in parietal VOIs did not correlate with neuropsychological performance. However, in frontal VOIs, there were significant correlations between SCWT time and Cho/Cr ratios (*r* = 0.32, *p* = 0.04), SDMT and mI/Cr (*r* = −0.37, *p* = 0.02) and NAA/mI (*r* = 0.39, *p* = 0.01) ratios, Go/no-go RT and NAA/Cho (*R* = −0.41, *p* = 0.009) and NAA/mI (*R* = −0.34, *p* = 0.03) ratios, and CVLT T5 and NAA/Cr ratios (*R* = 0.32, *p* = 0.047) (Figure 2). As SDMT score was associated with BDI/HADS, multiple linear regression models, including both ^1^H-MRS ratios and BDI or HADS scores for each subscale, were performed and the correlations between ^1^H-MRS ratios and SDMT scores remained significant. The applied Bonferroni correction confirmed the significance only for the correlation between Go/no-go RT and NAA/Cho.

## 4. Discussion

Cognitive impairment is one of the significant clinical manifestations of MS, as well as CIS. Structural, biochemical, and functional bases of cognitive impairment still require a thorough understanding and description. In this study, cognitive performance and affective symptoms were evaluated in the context of neurochemical parameters of NAWM via ^1^H-MRS in patients with CIS. The available literature states that the profile of cognitive deficits in CIS is similar to that of MS [1,4]. We found that patients with CIS performed worse in the tests involving attention, executive functions, working memory, and especially information processing speed, a deficit that was found previously [1,4]. Verbal memory was not impaired, which is also consistent with a previous study [33].

The pathological changes in the brain responsible for cognitive dysfunction in patients with MS and CIS are complex, including inflammatory and neurodegenerative changes affecting both gray and white matter (globally and regionally). Cognitive status is also affected by functional changes. Thus, it is difficult to find a single marker that highly correlates with cognitive performance, especially in patients at the early stage of disease. Among the classic MRI parameters, measures of atrophy, including the width of the third ventricle [34], are most associated with cognitive deterioration in patients diagnosed with MS [1,5,28]. However, volumetric measurements are not associated with cognitive impairment in patients with CIS [1,4,5,6]. In the present study, we used the width of the third ventricle as the only measure of atrophy, which was not associated with cognitive performance in patients with CIS. However, the presence of gadolinium-enhanced lesions correlated with errors in the Go/no-go task. These lesions are more common in MS patients with cognitive symptoms as the predominant manifestation [35], but another study indicated that the presence of these lesions is not associated with cognitive impairment [36].

Notably, several ^1^H-MRS measurements were associated with cognitive performance in patients with CIS in this study. ^1^H-MRS reveals damage to the NABT in patients with MS and CIS [8,10]. Although there are neurochemical parameters associated with cognitive function in patients with MS [10,11], it is not known whether ^1^H-MRS outcomes are predictive of cognitive impairment in patients with CIS. One study assessing magnetization transfer ratios in cortex found that they independently predicted cognitive impairment in patients with CIS [1]. A similar relationship was observed with abnormal fractional anisotropy (FA) in the prefrontal cortex [7], but not in the NAWM of the corpus callosum.

As we did not perform ^1^H-MRS on the HPs in our study, we were not able to compare their metabolite ratios with those of the CIS group. However, patients with CIS were previously shown to have lower NAA/Cr and NAA/Cho ratios than control participants [8], as well as increased mI levels [9]. NAA levels are considered to be a marker of axonal integrity and neuronal density, whereas mI levels reflect glial cell proliferation and activity; Cho levels reflect membrane disturbances, myelin damage, and repair processes [16]. We found that these measures, indicative of damage to NAWM, were related to the early cognitive deterioration in patients with CIS. Specifically, psychomotor speed, attention, and executive function were associated with mI/Cr, NAA/mI, Cho/Cr, and NAA/Cho ratios. NAA/Cr ratio, corresponding with neural and axonal density, was related only to verbal memory. It was shown that reduced NAA and Cho concentrations in the frontal white matter are associated with verbal memory impairment in MS patients [10]. Our findings are partially compliant with this observation as we found an association between NAA/Cr and verbal memory, but such correlations for Cho/Cr or NAA/Cho were not observed. Moreover, in the abovementioned study, frontal VOI was localized similar to that of our protocol. Other studies analyzing associations between ^1^H-MRS parameters within variously localized VOIs and cognitive functions in patients with MS revealed that decreased NAA/Cr and NAA/Cho ratios in the posterior periventricular region of NAWM correlated with impaired psychomotor speed, visuospatial ability, and verbal fluency; NAA/Cr ratios in the normal-appearing gray matter of the cingulate gyrus correlated with verbal memory; and NAA/Cr ratios in the locus coeruleus correlated with attention deficits [11,37,38]. These studies did not assess mI or mI/Cr ratios.

As the associations found in the present study were weak or moderate at most, damage to the NAWM is clearly only one pathophysiological factor affecting cognitive function in patients with CIS. Damage to the cortex, not evaluated in this study, is another factor that likely influences cognitive performance [1,7]. Therefore, a combination of factors remains superior to any single parameter for predicting cognitive impairment, as was previously indicated in patients with MS [11].

Only metabolic ratios in the frontal VOIs were associated with cognitive performance in this study. The frontal lobes, especially prefrontal regions, are essential for executive functions and working memory [39]. SCWT and Go/no-go task involve working memory and cognitive inhibition, which are strictly associated with executive functions; SDMT also involves working memory. Verbal memory is associated with the frontal lobes functioning as well [40]. Therefore, correlations between performance on these tests and damage to the frontal regions are understandable. Damage to NABT in frontal regions [10] but not in parietal regions [41] is associated with cognitive impairment in patients with MS. Accordingly, we observed lower NAA/Cr and NAA/Cho ratios in the frontal VOIs than in the parietal VOIs, but we are not able to state whether this is a consequence of disease or a normal anatomical variation as ^1^H-MRS study was not conducted in healthy participants. However, previous studies suggest that there is no region difference of metabolites concentrations in the white matter of healthy brains [42] or such a difference was shown only for Cho concentration (lower concentration in the posterior regions of white matter compared to anterior regions) [43]. Thus, the lower NAA/Cr ratio in the frontal VOIs than in the parietal VOIs observed in our study might suggest that the frontal regions of white matter are more vulnerable to damage in patients with CIS. Neurodegeneration in the frontal regions corresponds to the profile of cognitive deterioration observed in this study. This is consistent with other studies showing that frontotemporal regions exhibit the most atrophy and only the frontal cortex exhibits greater FA in patients with CIS [4,7]. Marked damage to the frontal regions was also noticed in the study on patients with MS [44]. A different pattern of damage to NAWM was observed in a study of Asian patients with MS, which revealed that the largest decrease in NAA was in parietal NAWM [45].

NAA/mI ratio was associated with central atrophy in this study. However, the association was weak, possibly because the atrophy is milder in patients with CIS. By contrast, diffuse brain pathology correlates with brain atrophy in patients with MS [46]. A possible reason why the width of the third ventricle was associated only with the NAA/mI ratio is higher sensitivity of the NAA/mI ratio to detect brain tissue damage than NAA/Cr or mI/Cr, as this combined ratio indicates both axonal density and glial cell activity. In the previous paragraph, we suggested that frontal regions might be more vulnerable to damage in patients with CIS. This could also explain why frontal VOIs correlated with the central atrophy, whereas parietal VOIs did not.

Affective symptoms are more frequent in patients with CIS than in the healthy population, with signs of depression and anxiety observed in up to 34% of patients [4,12]. Consistent with this, 22.5% and 35% of the patients in the present study had depression and anxiety symptoms, respectively. Previous studies suggested that depression in patients with CIS does not affect cognitive performance [2,12]. However, another study indicated that depression symptoms are associated with slower information processing [14]. Our results are in line with this as SDMT performance was associated with symptoms of both depression and anxiety. Moreover, executive function and verbal memory were associated with higher scores of depression.

Associations between affective symptoms and MRI findings in patients with MS and CIS have been noted. For example, right temporal, left insular, and right occipital lesion loads are linked to depression symptoms and white matter volume is related to anxiety in patients with CIS [14]. In patients with MS, symptoms of depression correlate with atrophy of cortical and subcortical gray matter [13]. The width of the third ventricle also correlates with depression symptoms in patients with MS [47] and a similar trend was observed in the patients with CIS in the present study, but the correlation did not reach statistical significance. It was shown that depression symptoms are associated with damage to normal-appearing white and gray matter in frontal and temporal regions assessed with DTI in patients with MS [15]. Consistent with this, neurochemical changes in the NAWM were related to affective symptoms in patients with CIS in the present study. Metabolic ratios in both frontal and parietal VOIs were associated with signs of depression, whereas only NAA/Cr in parietal VOIs correlated with anxiety symptoms. It is possible that damage to NAWM alters the connectivity between cortical and limbic regions, which results in depression and anxiety symptoms observed in the patients in the present study. Particularly the frontal lobes, but also parietal limbic-cortical pathways, are implicated in depressive symptoms. The role of sensory-related brain regions, such as parietal lobes, for anxiety symptoms was also emphasized [48,49].

Notably, we did not evaluate absolute concentrations of metabolites, but their amounts relative to Cr. Although a post-mortem study showed that the concentration of Cr is unchanged in the NABT of patients with MS [50], another study found increased levels of Cr in the NAWM in these patients [10]. Thus, the use of ratios to Cr levels may introduce error. However, NAA/mI ratios in MS have been emphasized [17], representing sensitive detection of damage to brain tissue by indicating both axonal integrity and glial cell activity. The NAA/Cho ratio further increases the sensitivity to detect subtle damage to brain tissue [18]. The strongest associations between ^1^H-MRS outcomes and cognitive performance in the present study were for NAA/mI and NAA/Cho. Thus, sensitive methods are crucial for evaluating the milder abnormalities that occur at early stages of disease. A major limitation of the study is the absence of MRI for HPs, precluding any comparisons of ^1^H-MRS outcomes between the groups. Other limitations include the relatively small group sizes and evaluation of width of the third ventricle as the only measure of atrophy. We only selected two VOIs in the NAWM for analysis of focal rather than of more diffuse damage. Other regions of the brain are involved in cognitive processes, including temporal lobes, which are associated with verbal memory, and the cerebellum, which is involved in information processing speed. These cognitive domains were assessed in this study, whereas ^1^H-MRS of the cerebellum and temporal lobe was not performed. Visual memory, which is affected frequently in CIS, was not evaluated in the study; also, subjects were not checked for fatigue. Additionally, premorbid intelligence was not checked in patients with CIS to compare it with HPs.

## 5. Conclusions

We demonstrated that neurochemical parameters of the NAWM via single-voxel ^1^H-MRS partially explains the variability of cognitive performance and affective symptoms in patients with CIS. Notably, metabolite ratios of NAWM in frontal regions were associated with cognitive functioning, whereas width of the third ventricle was not. Therefore, unconventional MRI techniques should be explored further to assess the impact of diffuse brain pathology on cognitive deterioration and other clinical outcomes in patients with CIS.

## Figures and Tables

**Figure 1 jcm-09-03909-f001:**
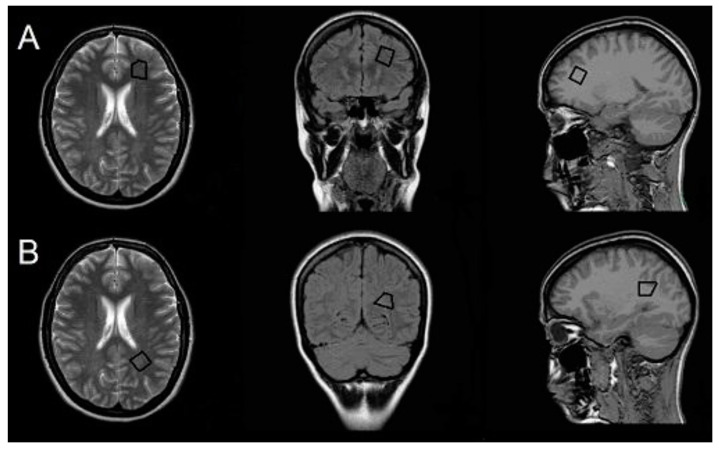
Volume of interest. Positions of volumes of interest (VOIs) in normal-appearing white matter (NAWM) in frontal (**A**) and parietal (**B**) regions.

**Figure 2 jcm-09-03909-f002:**
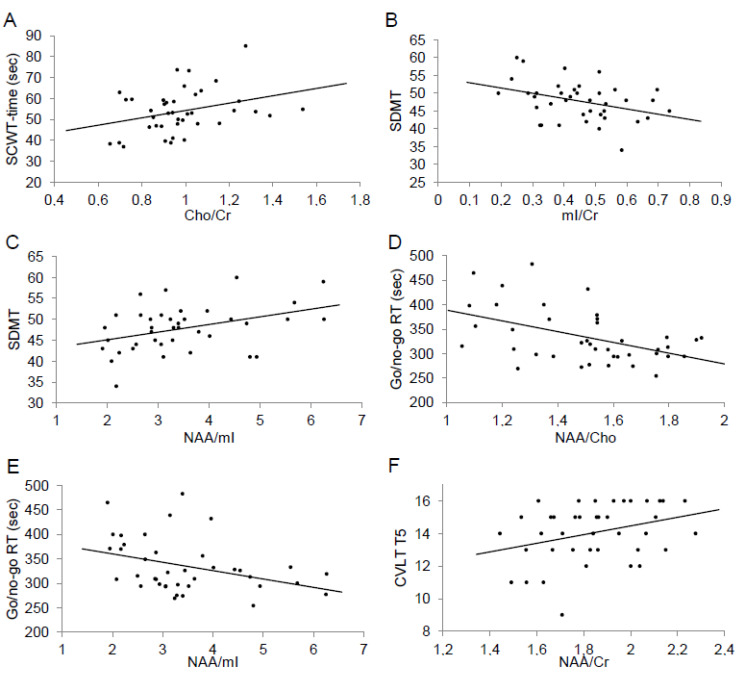
(**A**–**F**) Correlations between neuropsychological test scores and ^1^H-MRS ratios in frontal VOIs.

**Table 1 jcm-09-03909-t001:** Demographic and clinical characteristics of the study population.

Characteristic	CIS Group (*n* = 40)	HP Group (*n* = 40)	*p*-Value
Age (years)	32 (26.5–38.0)	30 (26.5–39.5)	0.92
Female-to-male ratio	29:11	28:12	0.8
Education (years)	15 (12–17)	16 (13–17)	0.37
Disease duration (months)	5 (3–8)		
CIS type			
Optic neuritis	19 (47.5)		
Transverse myelitis	9 (22.5)		
Brainstem/cerebellar	9 (22.5)		
Cerebral	3 (7.5)		
EDSS	1.25 (1.0–1.5)		

Values are presented as medians (25–75th percentiles) or n (%). CIS, clinically isolated syndrome; HP, healthy participant; EDSS, Expanded Disability Status Scale. Mann–Whitney U-test was used to assess differences in age and education, whereas a chi-square test was used to assess female-to-male ratios.

**Table 2 jcm-09-03909-t002:** Differences in cognitive test results and BDI and HADS in CIS and HP groups.

Test	CIS Group (*n* = 40)	HP Group (*n* = 40)	Effect Size	*p*-Value
TMT A (s)	29.87 ± 10.45	26.82 ± 7.06	0.43	0.13
TMT B (s)	64.50 (45.85–76.70)	50.95 (44.05–61.35)	0.56	0.02
SCWT time (s)	53.10 (47.35–59.30)	46.90 (39.50–55.10)	0.51	0.03
SCWT errors	0 (0–1.00)	0 (0–1.00)	0.13	0.57
SDMT	47.88 ± 5.1	55.50 ± 7.15	1.07	<0.0001
Verbal Fluency Test	31.0 (26.0–36.0)	33.5 (28.0–33.5)	0.18	0.43
SRT (s)	261.5 (236.0–288.5)	247.0 (220.0–280.5)	0.22	0.32
Go/no-go RT (s)	317.0 (294.0–366.5)	299.5 (278.5–341.5)	0.35	0.12
No-go errors	3.0 (1.5–4.5)	3.0 (2.0–5.5)	0.21	0.35
CVLT TT	60.30 ± 7.07	62.90 ± 5.42	0.41	0.07
CVLT T1	9.0 (8.0–10.0)	9.0 (8.0–10.0)	0.14	0.55
CVLT T5	14,0 (13.0–15.5)	15.0 (14.0–16.0)	0.25	0.27
CVLT SDR	13.0 (11.0–14.0)	14.0 (13.0–15.0)	0.50	0.04
CVLT LDR	13.5 (12.5–15.0)	14.0 (13.0–15.0)	0.23	0.32
BDI	5.0 (3.0–11.5)	3 (1–7)	0.51	0.03
HADS-D	2.0 (1.0–4.5)	1.0 (0.5–4.0)	0.36	0.12
HADS-A	5.5 (4.0–8.0)	3.5 (1.5–6.5)	0.65	0.006

Data are presented as means ± SDs or medians (25–75th percentiles). TMT A, Trail Making Test part A; TMT B, Trail Making Test part B; SCWT, Stroop Color-Word Test; SDMT, Symbol Digit Modalities Test; SRT, simple reaction time; RT, reaction time; CVLT, Californian Verbal Learning Test; TT, total trials; T1, trial 1; T5, trial 5; SDR, short delayed free recall; LDR, long delayed free recall; BDI, Beck Depression Inventory; HADS, Hospital Anxiety and Depression Scale; HADS-D, depression subscale; HADS-A, anxiety subscale; CIS, clinically isolated syndrome; HP, healthy participants. Independent sample t-tests were used for variables with normal distributions (data are presented as means ± SDs); Mann–Whitney U-tests were used for variables without normal distributions (data are presented as medians with 25–75th percentiles); effect size, Cohen’s d for t-test is shown for CVLT TT, Glass’ Δ is shown for TMT A and SDMT (due to inhomogeneous variations in the groups), and Cohen’s d for Mann–Whitney U-test is shown for other cognitive tests.

**Table 3 jcm-09-03909-t003:** ^1^H-MRS ratios in the frontal and parietal VOIs.

^1^H-MRS Ratio	Frontal VOI	Parietal VOI	Effect Size	*p*-Value
NAA/Cr	1.85 (0.21)	2.05 (0.31)	0.75	<0.001
mI/Cr	0.45 (0.13)	0.46 (0.13)	0.13	0.45
Cho/Cr	0.98 (0.19)	0.92 (0.15)	0.32	0.05
NAA/Cho	1.50 (0.24)	1.78 (0.26)	1.08	<0.001
NAA/mI	3.48 (1.17)	3.80 (1.27)	0.25	0.1

Data are presented as means ± SDs. ^1^H-MRS, proton magnetic resonance spectroscopy; VOI, volume of interest; NAA, N-acetyl-aspartate; mI, myo-inositol; Cho, choline; Cr, creatine. Paired sample *t*-tests were used for statistical analysis; Cohen’s d is presented as a measure of the effect size.

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
