# Peer review of "Neurochemical Changes in the Brain and Neuropsychiatric Symptoms in Clinically Isolated Syndrome"

_jcm, 2020, doi:10.3390/jcm9123909_

Round 1

Reviewer 1 Report

The article "Neurochemical Changes in Brain and Neuropsychiatric Symptoms in Clinically Isolated Syndrome" by Guenter et al. addresses psychiatric and an cognitive symptoms in patients with CIS and correlation of these symptoms to structural and neurochemical changes, which is an important area receiving growing attention in the field of MS research.

However, there are several major points of criticism:

Patient selection: The authors do not state, which version of the McDonald criteria was chosen for selection of CIS patients. A scientific publication in 2020 should use the current McDonald criteria of 2017. however, regarding the details of the MRI investigations with two thirds of patients showing DIS and 20% having gadolinium-enhanced lesions, it seems unlikely that according to the current criteria, these patients qualify as CIS patients but rather already fulfill the criteria for MS. This adaption would change the overall direction and message of this manuscript. Regarding this point, intrathecal IgG synthesis should also be given, since it constitutes an important factor in the current diagnostic criteria.

MRI atrophy measurement: Although width of the third ventricle is one of different markers of brain atrophy, in recent years many different markers for regional- and whole-brain-atrophy have been established. If the authors want to address atrophy in their analysis, additional parameters should be chosen.

Spectroscopy: The fact that spectroscopy was only conducted in the MS patients and not in the control group as well, considerably limits the significance of the work. Thus, the conclusions which the authors draw from their results, are not valid.

Author Response

Dear Reviewer,

Thank you for your valuable comments and feedback regarding our research. We have made some changes to the manuscript, as well as recorded the answers to the comments below.

  • Patient selection: The authors do not state, which version of the McDonald criteria was chosen for selection of CIS patients. A scientific publication in 2020 should use the current McDonald criteria of 2017. however, regarding the details of the MRI investigations with two thirds of patients showing DIS and 20% having gadolinium-enhanced lesions, it seems unlikely that according to the current criteria, these patients qualify as CIS patients but rather already fulfill the criteria for MS. This adaption would change the overall direction and message of this manuscript. Regarding this point, intrathecal IgG synthesis should also be given, since it constitutes an important factor in the current diagnostic criteria.

The study was prospective. While the study has been conducting, McDonald 2010 criteria were the applicable criteria for MS. Intrathecal IgG synthesis was not included in this criteria. CIS definition according to Miller et. al. 2012 was used. CIS refers to a first clinical CNS demyelinating event lasting ≥24 hours. We recruited patients as it is indicated in lines 60-69. Obviously, some patients after a first clinical CNS demyelinating event fulfill the criteria for MS based on MRI examination. A new revision by MAGNIMS of the McDonald 2010 criteria were published in 2016, so we retrospectively used this revised criteria to define which study participants fulfilled the criteria for MS. It is indicated in the Results section (lines 235-237), that eight patients fulfilled MAGNIMS 2016 criteria for MS. McDonald 2017 criteria could not be aplied, because lumbar puncture was not included in the study protocol.

Some comments were applied to the manuscript (lines 60 and 69-71).

  • MRI atrophy measurement: Although width of the third ventricle is one of different markers of brain atrophy, in recent years many different markers for regional- and whole-brain-atrophy have been established. If the authors want to address atrophy in their analysis, additional parameters should be chosen.

We included in the limitations of the study, that the only measure of atrophy was width of the third ventricle, as a measure of the central atrophy (lines 439-440). Width of the third ventricle is a simple parameter, which can be widely used in the clinical practice. It is currently impossible to assess additional parameters of atrophy in the study group.

  • Spectroscopy: The fact that spectroscopy was only conducted in the MS patients and not in the control group as well, considerably limits the significance of the work. Thus, the conclusions which the authors draw from their results, are not valid.

This problem is emphasized in lines 347-348 and in the limitations of the study (lines 436-437). However, patients with CIS were previously shown to have neurochemical changes in the normal-appearing brain tissue via MRS (lines 348-350). Obviously, results would be more informative, if MRS was conducted in the control group. Therefore, we modified conclusions (lines 448-451).

Reviewer 2 Report

This manuscript examined the correlation between MRS metabolite ratios and neuropsychological test scores in CIS. Significant correlations were found between a number of tests and different ratios but only in the frontal VOI. The metabolite ratios were better able to explain cognitive impairment than structural brain changes.

Expanding the relationship between metabolite concentration and cognition into the earliest phase of MS (namely CIS) is interesting and important. I was, however, disappointed that the authors only used ratios for their MRS measurements. As mentioned in the discussion, using a ratio means that you do not know whether the numerator or the denominator is changing. Although the ratio may be sensitive to change, the measurement will be less specific. In this day and age, individual metabolite quantification should be done routinely and ratios could be added as wanted.

Specific comments:

  1. A lot of statistical comparisons are being done so should correction for multiple comparisons not be applied to the p-value?
  2. Lesions in the cervical spine were mentioned in the results although I saw nothing about the spine being imaged in the methods.
  3. Patients with DIS were not different from the rest but how about the 8 patients that fulfilled the new diagnosis of MS criteria? Were they worse in their test scores or MRS concentrations?
  4. It would be nice to have the r/R and p-values on Figure 2 plots rather than having to find them in the text.
  5. More detail is needed for the multiple linear regression models. I was unsure what was being added to the model. Was this used to validate the statement “neurochemical assessment, including NAA/mI and NAA/Cho ratios, is more important for explaining cognitive impairment in patients with CIS than structural brain changes observed via conventional MRI”? Otherwise, I’m not sure how you can say that one measurement is better than the other.
  6. I was not sure of the purpose of including the controls in this study. It seems that their role was to be able to compare cognitive test scores but they weren’t part of the MRS study which was the main point of the paper. In the discussion, it’s mentioned that CIS have previously shown differences in metabolite ratios compared to controls. However, comparison between CIS and controls has also previously been done for cognitive tests. Were there some novel tests included?
  7. In the discussion (p.8, paragraph starting at line 281), the significant associations between ratios and cognitive tests are mentioned. Then follows a list for MS however the ROIs for MS are not in frontal areas so I’m not sure how comparable they are.
  8. In the discussion, it’s mentioned that “Damage to NABT in frontal regions (10) but not in parietal regions (35) is associated with cognitive impairment in patients with MS. Accordingly, we observed lower NAA/Cr and NAA/Cho ratios in the frontal VOIs than in the parietal VOIs.” But, is this a consequence of disease or is this a normal anatomical variation?
  9. In discussion, it is stated that there are no differences in metabolite concentrations within healthy white matter. I was surprised by this. The reference listed is older (2001) and newer references (e.g. Baker et al. “Regional Apparent Metabolite Concentrations in Young Adult Brain Measured by 1H MR Spectroscopy at 3 Tesla: JMRI, 2008; 27:489-99) state that there are differences in concentrations around the brain white matter.

What motivated the choice of VOIs? I understand the choice of frontal ROI but not necessarily the parietal ROI.

Author Response

Dear Reviewer,

Thank you for your valuable comments and feedback regarding our research. We have made some changes to the manuscript, as well as recorded the answers to the comments below.

  1. A lot of statistical comparisons are being done so should correction for multiple comparisons not be applied to the p-value?

The description of the multiple testing method used has been added in the method section (lines 187-189), and its specific applications in the corresponding result descriptions (lines 202-204, 205-206, 283-285, 294-295).

  1. Lesions in the cervical spine were mentioned in the results although I saw nothing about the spine being imaged in the methods.

The cervical spine was imaged. We included information about spine MRI in Methods section (lines 132-137).

  1. Patients with DIS were not different from the rest but how about the 8 patients that fulfilled the new diagnosis of MS criteria? Were they worse in their test scores or MRS concentrations?

The eight patients that fulfilled diagnostics criteria of MS were exactly the same patients who had gadolinium-enhanced lesions (information added in lines 239-240), so they made more no-go mistakes in the Go/no-go task (lines 247-248) and had lower parietal NAA/Cho ratio (lines 274-275)

  1. It would be nice to have the r/R and p-values on Figure 2 plots rather than having to find them in the text.

r/R and p-values were added to Figure 2.

  1. More detail is needed for the multiple linear regression models. I was unsure what was being added to the model. Was this used to validate the statement “neurochemical assessment, including NAA/mI and NAA/Cho ratios, is more important for explaining cognitive impairment in patients with CIS than structural brain changes observed via conventional MRI”? Otherwise, I’m not sure how you can say that one measurement is better than the other.

Multiple linear regression model mentioned in lines 291-294 included 1H-MRS ratios and BDI/HADS, to rule out influence of affective symptoms on associations between mI/Cr and SDMT as well as between NAA/mI and SDMT. It was performed, because SDMT was associated with affective symptoms in the simple regression.

We did not performed multiple linear regression model which included width of the third ventricle. Conclusion was modified, as 1H-MRS ratios correlated with cognitive functioning, whereas width of the third ventricle did not. We removed the statement, that one measurement is better than the other (lines 450-455).

  1. I was not sure of the purpose of including the controls in this study. It seems that their role was to be able to compare cognitive test scores but they weren’t part of the MRS study which was the main point of the paper. In the discussion, it’s mentioned that CIS have previously shown differences in metabolite ratios compared to controls. However, comparison between CIS and controls has also previously been done for cognitive tests. Were there some novel tests included?

The only role of the control group was to compare cognitive and affective test scores. We used the battery of neuropsychological tests which has not been used before. Especially, simple reaction time and go/no-go task are not included in the standard sets of neuropchychological tools for CIS/MS patients. Lilimations resulting from the fact, that MRS was not conducted in the control group are emphasized in lines 347-348 and 436-437.

  1. In the discussion (p.8, paragraph starting at line 281), the significant associations between ratios and cognitive tests are mentioned. Then follows a list for MS however the ROIs for MS are not in frontal areas so I’m not sure how comparable they are.

This part of discussion was modified (lines 354-367).

  1. In the discussion, it’s mentioned that “Damage to NABT in frontal regions (10) but not in parietal regions (35) is associated with cognitive impairment in patients with MS. Accordingly, we observed lower NAA/Cr and NAA/Cho ratios in the frontal VOIs than in the parietal VOIs.” But, is this a consequence of disease or is this a normal anatomical variation?

This comment was added to discussion (lines 380-382).

  1. In discussion, it is stated that there are no differences in metabolite concentrations within healthy white matter. I was surprised by this. The reference listed is older (2001) and newer references (e.g. Baker et al. “Regional Apparent Metabolite Concentrations in Young Adult Brain Measured by 1H MR Spectroscopy at 3 Tesla: JMRI, 2008; 27:489-99) state that there are differences in concentrations around the brain white matter.

According to Baker et al. (2008) there is no region difference of metabolites concentration in the white matter of healthy brains, with exception for Cho. Concentration of Cho is lower in the posterior regions of white matter compared to anterior regions (Table 2, p. 17-18). It could explain lower NAA/Cho ratio in frontal VOI of CIS patients, but it still do not explain why NAA/Cr was decreased in frontal VOI. MRS was not conducted in healthy participants in our study, so we are not able to state that neurodegeneration is more pronounced in the frontal regions of white matter in CIS patients. However, we think it is some suggestion indicating the need for further investigation. We included this comment (lines 383-388).

What motivated the choice of VOIs? I understand the choice of frontal ROI but not necessarily the parietal ROI.

We presumed that correlations between metabolic ratios and neuropsychological performance might be more pronounced in frontal VOI. Especially since most of the neuropsychological tests we administered involve executive functions. Thus parietal VOI was taken as a control one (regarding cognitive functions).

Reviewer 3 Report

This work on "Neurochemical changes in brain and neuropsychiatric symptoms in clinically isolated syndrome" investigates associations between cognitive impairment and affective symptoms and the biochemical profile in the brain of patients with clinically isolated syndrome (CIS). Neuropsychological testing and 1H magnetic resonance spectroscopy were conducted to collect the target measures. Some correlations were found between cognitive function and symptoms and MRS peak ratios.

MAJOR COMMENTS:
- Overall the work is quite interesting. Finding imaging markers and predictors of disease progression at an early stage is an active area of MS research. The study is generally well-conducted, and the manuscript is well-written. There are nevertheless some concerns with this otherwise nice work. First, the statistical analysis does not appear to account for the many multiple comparisons made in this study. This can potentially lead to many false discoveries. I wonder how many correlations will stand a correction for multiple comparisons. Second, as the authors acknowledge, no MRS was done in the control group (healthy participants), which limit the strength of the metabolic findings. Third, there is not enough discussion of the identified significant correlations. Are these correlations with MRS in line with the symptoms/CI?

ABSTRACT:
- Please mention the statistical test used.

METHODS
- I wonder why the authors have not tried absolute water-referenced metabolite quantification rather than resorting to metabolite ratios.

- Could the authors comment on why they mix results obtained at both short- and long-TE MRS. The peak ratios
are expected to be strongly affected by TE.

- With so many correlations assessed, it is not clear why adjustment for multiple comparisons was not made.

DISCUSSION
-It would be instructive if the authors try to explain some of the results. For example: "Metabolic ratios in both frontal and parietal VOIs were associated with signs of depression, whereas only NAA/Cr in parietal VOIs correlated with anxiety symptoms".

- Could the authors comment on why the width of the third ventricle was correlated only with the frontal NAA/mI ratio?

OTHER COMMENTS:
- Please describe what technique was used for water suppression in 1H-MRS.
- Please be consistent using ‘transverse’ or ‘axial’ when describing slice orientation.
- Please provide a reference for Barkhof/Tintoré MRI criteria for DIS.

Author Response

Dear Reviewer,

Thank you for your valuable comments and feedback regarding our research. We have made some changes to the manuscript, as well as recorded the answers to the comments below.

ABSTRACT:
- Please mention the statistical test used.

Unfortunately, even the shortest description of statistical methods will result in exceeding the limit of words in the abstract imposed by the publisher. Hence, it is only possible with the publisher's consent to extend the abstract by another several dozen words.

METHODS
- I wonder why the authors have not tried absolute water-referenced metabolite quantification rather than resorting to metabolite ratios.

The study was planned in 2014 and conducted in 2014-2016. Currently, we are not able to change or improve methods. This issue is discussed in lines 427-430 as a limitation. However, as we mentioned, there are some advantages of using ratios, especially combined ratios, as NAA/mI and NAA/Cho.

- Could the authors comment on why they mix results obtained at both short- and long-TE MRS. The peak ratios
are expected to be strongly affected by TE.

NAA/Cr was measured at long TE due to possible artifactual elevation of NAA in short TE. Cho/Cr and mI/Cr were assessed at short TE due to non-detection of short T1/T2 metabolites such as mI and artifactual elevation of Cho at long TE. We assessed NAA/Cho ratio at long TE, because the effects of T2 decay on the NAA and Cho signal at long TE will be similar so Cho/NAA ratio will show less TE dependence. Assessment of NAA/mI ratio at short TE may results in overestimation, as NAA could be elevated due to overlap with Glx complex, but it is impossible to measure NAA/mI at long TE. Thus we did not mix short and long TE peaks within one ratio.

We presented study results including MRS ratios together, irrespective of TE, because it is specified in Methods which ratios were obtained at short- or long TE. If it is recommended, we can separate results with ratios obtained at short TE from those at long TE.

- With so many correlations assessed, it is not clear why adjustment for multiple comparisons was not made.

The description of the multiple testing method used has been added in the method section (lines 187-189), and its specific applications in the corresponding result descriptions (lines 202-204, 205-206, 283-285, 294-295).

DISCUSSION
-It would be instructive if the authors try to explain some of the results. For example: "Metabolic ratios in both frontal and parietal VOIs were associated with signs of depression, whereas only NAA/Cr in parietal VOIs correlated with anxiety symptoms".

  • Additional comment was included (lines 354-361).
  • Explanation why metabolic ratios in frontal VOIs were associated with neuropsychological performance was included (lines 373-378).
  • A short comment concerning depression and anxiety was included (lines 424-426).

- Could the authors comment on why the width of the third ventricle was correlated only with the frontal NAA/mI ratio?

The comment was included in the discussion (lines 397-402).

OTHER COMMENTS:
- Please describe what technique was used for water suppression in 1H-MRS.

Included in the Methods section (lines 157-158).

- Please be consistent using ‘transverse’ or ‘axial’ when describing slice orientation.

It was modified.

- Please provide a reference for Barkhof/Tintoré MRI criteria for DIS.

Provided (lines 237-238).

Reviewer 4 Report

It is an interesting and well conducted work, addressing important questions about the neural substrates of cognitive impairment in CIS.

Minor changes, mainly in the form of limitations,  which can be summarized to the following:

  1. Subjects with CIS and healthy controls have been not checked and compared for premorbid intelligence

Or/and cognitive reserve, and this might influence the results.

  1. Visual memory, which is affected frequently in CIS, was not evaluated.

  1. Also subjects were not checked for fatigue.

  1. A major limitation is that 1H-MRS was not performed on healthy subjects forming the control group, so comparisons with metabolite ratios measured in the CIS group was not possible.

Author Response

Dear Reviewer,

Thank you for your valuable comments and feedback regarding our research. We have made some changes to the manuscript, as well as recorded the answers to the comments below.

  1. Subjects with CIS and healthy controls have been not checked and compared for premorbid intelligence

Or/and cognitive reserve, and this might influence the results.

Incuded in the limitations (lines 445-446).

  1. Visual memory, which is affected frequently in CIS, was not evaluated.

Incuded in the limitations (lines 444-445)

  1. Also subjects were not checked for fatigue.

 Incuded in the limitations (line 445).

  1. A major limitation is that 1H-MRS was not performed on healthy subjects forming the control group, so comparisons with metabolite ratios measured in the CIS group was not possible.

This problem is emphasized in lines 347-348 and in the limitations of the study (lines 436-437). However, patients with CIS were previously shown to have neurochemical changes in the normal-appearing brain tissue via MRS (lines 348-350). Obviously, results would be more informative, if MRS was conducted in the control group.

Reviewer 5 Report

The authors in the present work (Guenter W, et. al.) study the neucrochemical changes in the normal-appearing white matter (NAWM) in 2 regions of brains of patients with CIS (compared to matched healthy controls) using MR-Spectroscopy. They compare their spectroscopy data to neuropsychological tests, aiming to find clinicopathological correlations that could characterize the cognitive performance and affective symptoms of CIS patients.

Their work is important for the field of MS-diagnostic and adds important data related to the pathology of NAWM in a very early stage of neuroinflammation. Their conclusion that neurochemical assessment of NAWM can explain beginning cognitive impairment is highly important and is translational to relevant preclinical data on axonal pathology in NAWM.

In order to improve the current manuscript, I have few major and minor constructive comments/recommendations, before its publication in Journal of Clinical Medicine.

More specifically:

Major comments:

  1. Introduction: the authors should include a paragraph between lines 47 and 48 that will introduce the utility and practical significance of MR-Spectroscopy measurements in their study. Please introduce here what do the measurements of NAA/Cr, NAA/Cho, Cho/Cr, mI/Cr and NAA/mI each mean for the underlying brain pathology (such data are provided for example within lines 284-294). How are these measurements interpreted in cases of pathology?
  2. The authors did not perform MR-Spectroscopy in healthy controls. This is a major problem of the study as it removes important information: how can the authors validate whether the measured NAA/Cr, NAA/Cho, Cho/Cr, mI/Cr and NAA/mI in CIS-patients are pathological and thus truly indicating of the neuropsychological changes in these patients? A reference to previous studies (lines 283-284) may alone not be enough. Can the authors measure the healthy controls?
  3. The authors report slower information processing in their CIS-cohort (as shown by SDMT). Slower information processing is neuroanatomically associated with cerebellar lesions as well. However, relevant MR-Spectroscopy of cerebellum is not included in the study and limits the study. As the study seems to be a retrospective one, I suppose this is not now possible. Though, such a limitation should be clearly discussed and taken into account in the discussion of findings. If the authors can provide such measurements, they should be included in the study.

Minor comments:

  1. lines 217-222: please add for each one of reported results a short statement about their clinical significance. For example, what does it mean a no-difference between DIS+ and DIS-negative patients regarding BDI, HADS and cognition? Similarly for Gd-enhanced lesions and width of 3rd ventricle. Please reform appropriately (e.g. We tested the hypothesis that.... . Collectively our data indicate that....)
  2. lines 234-250: similarly, the authors report a lot of data without connecting them to a possible relevant significance. In other words, the paragraph does not contain a clear testing hypothesis to follow. Please reform appropriately (e.g. We tested the hypothesis that.... . Collectively our data indicate that....).
  3. line 256: please replace "We noticed..." to "We found..."
  4. Please reintroduce shortly and clear the testing hypothesis in the beginning of the discussion. Lines 263-267 provide, for example, a nice input of why MR-Spectroscopy is a needed for finding markers for cognitive performance.
  5. Is this study a retrospective or prospective one? Please state it.

Author Response

Dear Reviewer,

Thank you for your valuable comments and feedback regarding our research. We have made some changes to the manuscript, as well as recorded the answers to the comments below.

  1. Introduction: the authors should include a paragraph between lines 47 and 48 that will introduce the utility and practical significance of MR-Spectroscopy measurements in their study. Please introduce here what do the measurements of NAA/Cr, NAA/Cho, Cho/Cr, mI/Cr and NAA/mI each mean for the underlying brain pathology (such data are provided for example within lines 284-294). How are these measurements interpreted in cases of pathology?

Included (lines 48-54).

  1. The authors did not perform MR-Spectroscopy in healthy controls. This is a major problem of the study as it removes important information: how can the authors validate whether the measured NAA/Cr, NAA/Cho, Cho/Cr, mI/Cr and NAA/mI in CIS-patients are pathological and thus truly indicating of the neuropsychological changes in these patients? A reference to previous studies (lines 283-284) may alone not be enough. Can the authors measure the healthy controls?

Currently we can not conduct MRS on healthy participants. This problem is emphasized in lines 347-348 and in the limitations of the study (lines 436-437). Obviously, results would be more informative, if MRS was conducted in the control group.

  1. The authors report slower information processing in their CIS-cohort (as shown by SDMT). Slower information processing is neuroanatomically associated with cerebellar lesions as well. However, relevant MR-Spectroscopy of cerebellum is not included in the study and limits the study. As the study seems to be a retrospective one, I suppose this is not now possible. Though, such a limitation should be clearly discussed and taken into account in the discussion of findings. If the authors can provide such measurements, they should be included in the study.

We can not provide such measurements. Therefore it was add to the limitations (lines 441-444).

 Minor comments:

  1. lines 217-222: please add for each one of reported results a short statement about their clinical significance. For example, what does it mean a no-difference between DIS+ and DIS-negative patients regarding BDI, HADS and cognition? Similarly for Gd-enhanced lesions and width of 3rd ventricle. Please reform appropriately (e.g. We tested the hypothesis that.... . Collectively our data indicate that....)

The proposed narrative ordering of the results was applied (lines 242-253).

  1. lines 234-250: similarly, the authors report a lot of data without connecting them to a possible relevant significance. In other words, the paragraph does not contain a clear testing hypothesis to follow. Please reform appropriately (e.g. We tested the hypothesis that.... . Collectively our data indicate that....).

As recommended, the presentation of results was improved (lines 271-295).

  1. line 256: please replace "We noticed..." to "We found..."

The necessary amendment has been applied (line 322).

  1. Please reintroduce shortly and clear the testing hypothesis in the beginning of the discussion. Lines 263-267 provide, for example, a nice input of why MR-Spectroscopy is a needed for finding markers for cognitive performance.

The beginning of the “Discussion” was modified, also including the purpose of the study (lines 317-320).

  1. Is this study a retrospective or prospective one? Please state it.

The study was prospective (line 62).